# A computational model for angular velocity integration in a locust heading circuit

Kathrin Pabst [1,2¤]*, Evripidis Gkanias[3], Barbara Webb[3], Uwe Homberg[2,4], Dominik Endres[1,2]

**1** Department of Psychology, Philipps-Universität Marburg, Marburg, Hesse, Germany, **2** Center for Mind, Brain and Behavior (CMBB), Philipps-Universität Marburg, Justus Liebig Universität Giessen, and Technische Universität Darmstadt, Hesse, Germany, **3** School of Informatics, University of Edinburgh, Edinburgh, Scotland, United Kingdom, **4** Department of Biology, Philipps-Universität Marburg, Marburg, Hesse, Germany

¤ Current address: Faculty of Health, York University, Toronto, ON, Canada
* kmpabst@yorku.ca

**Data Availability Statement:** All relevant code to generate the figures is publicly available via https://gitlab.uni-marburg.de/fb04/ag-endres/computational-model-locust-heading-circuit.

## Abstract

Accurate navigation often requires the maintenance of a robust internal estimate of heading relative to external surroundings. We present a model for angular velocity integration in a desert locust heading circuit, applying concepts from early theoretical work on heading circuits in mammals to a novel biological context in insects. In contrast to similar models proposed for the fruit fly, this circuit model uses a single 360° heading direction representation and is updated by neuromodulatory angular velocity inputs. Our computational model was implemented using steady-state firing rate neurons with dynamical synapses. The circuit connectivity was constrained by biological data, and remaining degrees of freedom were optimised with a machine learning approach to yield physiologically plausible neuron activities. We demonstrate that the integration of heading and angular velocity in this circuit is robust to noise. The heading signal can be effectively used as input to an existing insect goal-directed steering circuit, adapted for outbound locomotion in a steady direction that resembles locust migration. Our study supports the possibility that similar computations for orientation may be implemented differently in the neural hardware of the fruit fly and the locust.

## Author summary

In both fruit flies and locusts, a specific brain region shows an activity pattern that resembles a compass, with an activity peak moving across an array of neurons as the animal rotates through 360 degrees. However, some apparent differences in the properties of this pattern between the two species suggest there may be differences in how this internal compass is implemented. Here we focus on the locust brain, building a computational model that is based on observed neural connections and using machine learning to tune the system. Turning by the simulated locust provides modulatory input to the neural circuit that keeps activity in the array aligned to its heading direction. We simulate a migrating locust that tries to keep the same heading despite perturbations and show this circuit

**Funding:** This work was financially supported in part by Deutsche Forschungsgemeinschaft (https://www.dfg.de/en, HO 950/28-1 to UH and EN 1152/3-1 to DE), by Hessisches Ministerium für Wissenschaft und Kunst (https://www.theadaptivemind.de/, project "The Adaptive Mind" to DE), and by Horizon Europe European Innovation Council (https://research-and-innovation.ec.europa.eu/funding/funding-opportunities/funding-programmes-and-open-calls/horizon-europe_en, project "InsectNeuroNano", 101046790 to BW). The funders had no role in study design, data collection and analysis, decision to publish, or preparation of the manuscript.

**Competing interests:** The authors have declared that no competing interests exist.

can steer it back on course. Our model differs from existing models of the fruit fly compass, showing how similar computations could have different implementations in different species.

## Introduction

Various navigational strategies have evolved across diverse ecological contexts, many relying on a robust estimate of the animal's current heading direction [14–16]. Shared across species (including humans) [17, 18], these strategies likely stem from similar neuronal and computational foundations. Investigating orientation and its neural substrates in a model organism provides a gateway to uncovering general mechanisms of spatial cognition. With their impressive navigational abilities and suitability for both laboratory and field studies, insects emerge as excellent model organisms for investigating navigation [19].

The navigation centre of the insect brain is located in the central complex (CX) [20–23]. This brain region is a midline-spanning group of four major neuropils: the protocerebral bridge (PB), the upper (CBU) and lower (CBL) divisions of the central body (also known as the ellipsoid body (EB) and fan-shaped body (FB) in some species; see Table 1 for a comparison of terms and their abbreviations between the desert locust (*Schistocerca gregaria*) and homologues in the fruit fly (*Drosophila melanogaster*); and the paired noduli (NO) [21]. The PB, CBU, and CBL are compartmentalised into columns, and the CBU, CBL, and NO are stratified into layers [1, 20, 24]. These columns and layers are interconnected by tangential and columnar neurons following stereotypical projection patterns [1, 9]. Tangential neurons provide multimodal inputs from various brain regions to the CX [3, 8, 25–27], while columnar neurons connect columns between the different neuropils and serve as the principal output elements of the CX [11, 21]. This organization is highly conserved, and tight structure-function relationships reveal the biological implementation of vector-based algorithms in the CX [23].

**Table 1. Abbreviations for neuron types and brain regions in the desert locust (*Schistocerca gregaria*) and homologues in the fruit fly (*Drosophila melanogaster*).**

| *Schistocerca gregaria* | *Drosophila melanogaster* |
| --- | --- |
| Lower division of the central body (CBL) | Ellipsoid body (EB) |
| Upper division of the central body (CBU) | Fan-shaped body (FB) |
| Central complex (CX) | Central complex (CX) |
| CL1a-neurons [1] | E-PG-neurons [2, 3] |
| CL2-neurons [4] | P-EN-neurons [2, 3] |
| CPU1-neurons [1] | PFL1/3-neurons [3] |
| CPU2-neurons [1] | PFL2-neurons [3] |
| CPU4-neurons [1] | PFN-neurons [3] |
| CU-neurons [1] | FC2-neurons [5] |
| Nodulus (NO) | Nodulus (NO) |
| Lower unit of the nodulus (NOL) | Lower unit of the nodulus (NOL) |
| Protocerebral bridge (PB) | Protocerebral bridge (PB) |
| TB1/2-neurons [6] | Δ7-neurons [7] |
| TB7-neurons [8] | SpsP-neurons [3, 7, 9] |
| TL-neurons [4, 10] | ER-neurons [3] |
| TNL-neurons [11] | L-N-neurons [12], LNO- & GLNO-neurons [3, 13] |

Analogous to head direction cells [14] observed in mammals [28, 29], numerous CX-neurons respond to celestial cues and map the animal's heading direction relative to the angle of polarised skylight and the solar azimuth [11, 14, 30]. To track the animal's orientation robustly, these cells integrate partly redundant inputs from various modalities [22], including self-motion generated signals [2, 31], such as efference copies or optic flow. In the fruit fly, columnar E-PG-neurons form a comprehensive 360° compass within the EB, with calcium imaging revealing a single activity maximum or compass bump encoding the animal's heading direction [2, 31]. Projection schemes of E-PG-neurons yield one 360° compass representation in each hemisphere of the PB, shifted relative to each other by 22.5° [2, 3, 32]. The representation of space in the fruit fly CX exhibits variability between individuals and across contexts [30, 31].

In contrast, in the desert locust, intracellular recordings suggest a single 360° compass encoding across the entire width of the PB [6, 33, 34], and projection patterns of columnar neurons imply two intercalated 180° representations of space along the CBL [1, 35]. The data suggest that the compass topography is consistent across individual locusts. Fig 1A

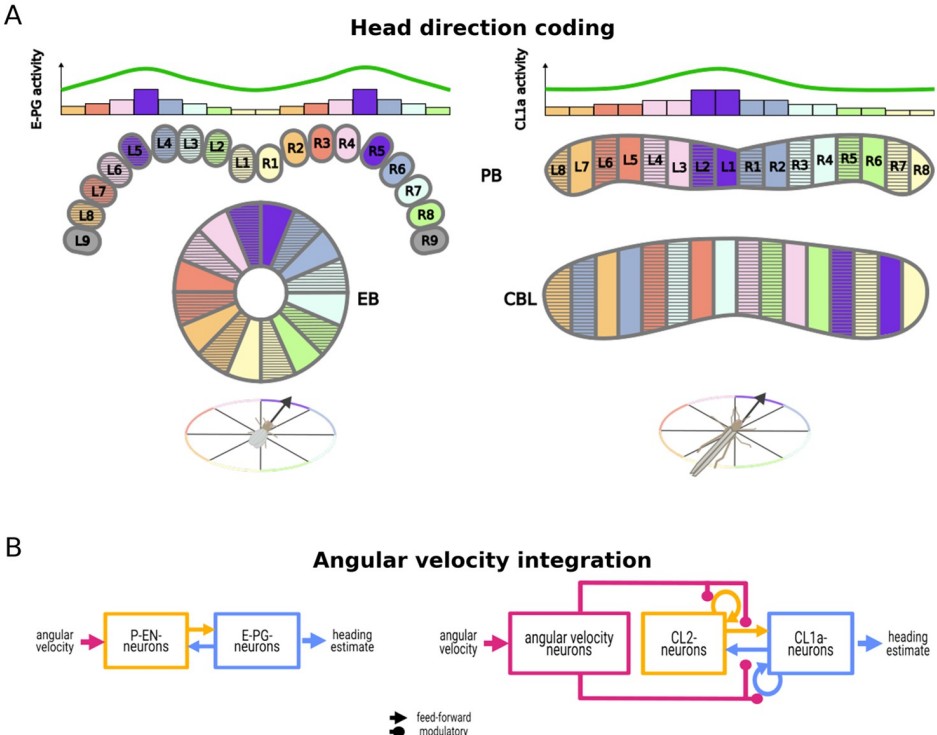

**Fig 1. Overview of the proposed model.** (A) Schematic comparison of heading encoding in the fruit fly and the desert locust protocerebral bridge (PB) and ellipsoid body / lower division of the central body (EB and CBL, respectively), based on data from [32, 42] and [35] and inspired by illustrations from [3, 23, 34]. Columnar E-PG- and CL1a-neurons encode the heading direction (indicated by color) of the insect. Bar graphs illustrate the activity level of neurons in each PB column, revealing a sinusoidal pattern of activity across the PB. In contrast to the fruit fly's 2 × 360° representation of space with two activity maxima (compass bumps) along the PB, one on either side, our model of the locust heading circuit assumes a 360° spatial map with a single compass bump along the entire PB. The EB features a single bump of activity, but projection patterns of CL1a-neurons [1] (cf. Fig 2A) imply two intercalated representations of space along the locust CBL [35]. (B) Diagrammatic comparison of information flow through the fruit fly heading circuit proposed by [2] and our proposed model of the desert locust heading circuit. Both circuits feature homologous columnar neurons (E-PG- and P-EN-neurons / CL1a- and CL2-neurons). In this fruit fly heading circuit, P-EN activity directly depends on the animal's angular velocity. In our proposed locust heading circuit, an abstract class of angular velocity neurons modulates the circuit connectivity depending on the animal's angular velocity.

illustrates the internal compass topographies in the PB and EB/CBL of the fruit fly and the desert locust, respectively. In addition to this functional difference between the fruit fly and locust heading circuits, many structural differences exist [36, 37]. A major difference is the ring-shaped EB in the fruit fly, which is a striking exception to the crescent-shaped homologous neuropils in most other insect species [38], including the CBL in the locust (cf. Fig 1A for the comparison of the fruit fly EB and desert locust CBL, respectively). Given the prevalence of this locust-like CBL architecture, we deemed it relevant to investigate the consequences of a locust-like $1 \times 360°$ heading representation in the PB. Many models of insect navigation [36, 39–41] assume a $2 \times 360°$ heading representation across the two halves of the PB, but this pattern might be an exception in the fruit fly rather than common to all insects.

A model of angular velocity integration in a fruit fly heading circuit [2] includes two types of columnar neurons, E-PG-neurons encoding heading and P-EN-neurons conjunctively encoding heading and angular velocity. Within this model, asymmetric activation of P-EN-neurons in the two halves of the PB occurs based on the fruit fly's turning direction, resulting in a shift of E-PG and P-EN activity maxima in the EB and the PB through the circuit's connectivity. This mechanism is consistent with an early theoretical framework of self-motion integration in a vertebrate heading circuit [43]. This framework proposed additive and multiplicative modulation that introduces asymmetries to the circuit connectivity as two alternatives to shift the compass bump. The model by [2] fits the additive modulation mechanism described by [43]. Recent studies have identified tangential GLNO-neurons as a source of rotational velocity information to P-EN-neurons in the fly. They rely primarily on motor signals but can alternatively use visual information [13]. In the locust, homologous columnar neurons to E-PG- and P-EN-neurons are CL1a- and CL2-neurons. They display projection schemes [1] suggesting similar connectivity, although excitation and inhibition remain uncertain. Functionally, CL1a-neurons encode heading relative to a fixed point of reference [33, 34], and recordings from CL2-neurons suggest directional sensitivity to rotational optic flow [35]. The circuit likely receives angular velocity inputs from tangential neurons [11]. Tangential neurons homologous to GLNO-neurons have been termed TNL-neurons [11], but data revealing their responsiveness to angular velocity are lacking.

In previous work, using a simplified model with linear neural units, discrete-time updates, and binary rotation encoding (left vs. right), we showed that the observed heading encoding in the locust could in principle be shifted appropriately by a multiplicative rotation-dependent modulation of the firing rate [35, 44]. Unlike the additive modulation mechanism observed in the fruit fly, our locust model aligns with the multiplicative modulation introduced by [43] and suggests a potential biological implementation in an insect heading circuit. Here, we significantly extend this work by developing a firing rate model of the locust heading circuit with synaptic dynamics and by optimising its function under structural and biologically plausible parameter constraints. An overview of the included neuron types and their interactions is shown in Figs 1B, and 2 features a more detailed depiction. We were interested in determining whether such a constrained model would be able to integrate a continuum of angular velocities and generate locust-like neural activity and orientation behaviour. This approach is conceptually related to the 'bounded rationality' models in cognitive science [45], where realistic behaviour emerges by training models constrained by available resources towards optimal behaviour. The results show that a heading circuit with a compass topography different from that in the fruit fly, and with neuromodulatory instead of feed-forward angular velocity inputs, can still function to maintain a robust heading estimate and to control steering behaviour.

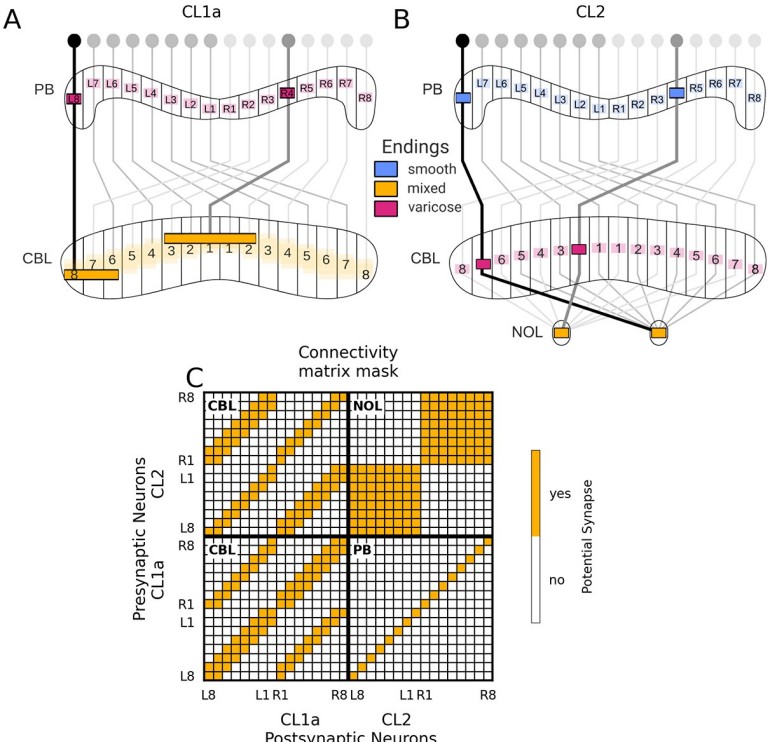

**Fig 2. Deduction of the heading circuit connectivity from anatomically plausible connections in the CX.** (A,B) Projection schemes of CL1a- and CL2-neurons, adapted from [1]. CL1a-neurons connect multiple adjacent columns of the CBL to single columns of the PB. CL2-neurons connect single columns of the PB to single columns of the CBL and to the contralateral NOL. The pattern by which PB and CBL are connected is shifted by one column when comparing the two neuron types. (C) Connectivity matrix mask for CL1a- and CL2-neurons, indicating potential synapses (yellow squares) without specifying excitation versus inhibition or strength. Neurons are arranged according to their position in the PB. PB, protocerebral bridge; CBL and CBU, lower and upper division of the central body, NOL, lower unit of the noduli.

## Model and methods

The heading circuit model and all simulations were implemented in Python 3.11. We used the machine learning library PyTorch [46] (version 2.2.1) for optimisation of the model's free parameters as described in section Free parameters and their optimisation.

### Neuron model

The network consists of single-compartment steady-state firing rate neurons abstracted from integrate-and-fire neurons [47]. Since the main focus of our work is the circuit topology, we only present the key assumptions regarding free and constrained parameters here. For a complete derivation of the neuron model, please refer to the supporting information S1 Text.

The dynamics of the model neurons are governed by two time constants: $\tau_m$, the membrane time constant, and $\tau_s$, the synaptic time constant. Estimates for $\tau_m$ vary with neuron type. While $\tau_m$ = 1.5 ms can be derived from the resting state conductance of a Hodgkin-Huxley model with the parameter values given in [47], the membrane time constants used in the fruit fly CX model of [2] are larger by an order of magnitude to capture observed delays between E-PG- and P-EN-neuron activity in walking flies [2]. Please refer to supporting information S2 Text for an order-of-magnitude delay estimation approach. We chose to model these delays

by slow synapses, i.e. $\tau_s \gg \tau_m$. This assumption justifies the use of a steady-state firing rate model with an explicit dynamics model for the synapses. In our computational model, "steady-state firing rate neurons" refers to a simplified representation where neurons are considered to achieve a stable firing rate after initial transients have settled. This stable firing rate is reached due to the slow dynamics of the synaptic inputs relative to the membrane time constant. Although the firing rate becomes stable after some time, it is still responsive to ongoing inputs and can change depending on the input dynamics. The steady-state potential $U_\infty$ of the membrane, which evolves on the same slow time scale as the neuron's inputs, is

$$U_\infty = \frac{\sum_i \tilde{g}_{tot,i} P_{s,i} E_{s,i}}{1 + \sum_i \tilde{g}_{tot,i} P_{s,i}}, \tag{1}$$

where $P_{s,i}$ is the post-synaptic ion channel opening probability, $E_{s,i}$ is the synaptic reversal potential, and $\tilde{g}_{tot,i}$ is the total relative synaptic conductance of synapse $i$. This conductance is calculated as

$$\tilde{g}_{tot,i} = \tilde{g}_{s,i} + \sum_j P_{rel,i,j} \tilde{g}_{mod,i,j}, \tag{2}$$

where $\tilde{g}_{mod,i,j}$ and $\tilde{g}_{s,i}$ are its two contributions. Note that $\tilde{g}_{mod,i,j}$ is multiplicatively modulated by pre-synaptic transmitter release probability $P_{rel,i,j}$ from modulatory input $j$, and $\tilde{g}_{s,i}$ is not modulated. Both contributions are free parameters, hereafter referred to as synaptic "weights", that will be optimized subject to connectivity constraints. In our model, the sign of the synaptic weight determines whether a synapse is excitatory or inhibitory (positive or negative synaptic weight, respectively). We compute the firing rate $r$ with a logistic sigmoid activation function

$$r(U_\infty) \approx \frac{99.6\,\text{Hz}}{1 + \exp(-0.19\,\text{mV}^{-1} \cdot (U_\infty - 17.8\,\text{mV}))}. \tag{3}$$

We model the effect of a spike on $P_{s,i}$ by a single-exponential kernel. The kernel is convolved with the density of the pre-synaptic spike train, which we assume to have inhomogeneous Poisson process statistics with rate $r_i(t)$. We argue that the Poisson assumption is approximately valid for CX neurons, since $r_i(t)$ does not exceed 50 Hz in available data from the locust [35]. This implies that the typical inter-spike interval is substantially longer than the refractory period. Following the derivation in [47], the time course of $P_{s,i}$ can then be described by the differential equation

$$\frac{dP_{s,i}}{dt} = P_{s,max} r_i(t) - \left(r_i(t) + \tau_s^{-1}\right) P_{s,i}, \tag{4}$$

where $P_{s,max}$ is the maximum synaptic open probability, which we set to 1.

The circuit model relies on multiplicative neuromodulation for heading representation updates. Neurons with a modulatory effect change their pre-synaptic transmitter release probability $P_{rel,i,j}$ proportionally to their rate and with time constant $\tau_{s,mod}$. We expect $\tau_{s,mod} > \tau_s$ because neuromodulation often involves signal transmission cascades.

## Heading circuit model

The heading circuit consists of columnar CL1a- and CL2-neurons and an abstract class of angular velocity neurons that combine properties of various tangential neurons. The angular velocity neurons are sensitive to angular velocity and modulate the connectivity among CL1a- and CL2-neurons.

The locust CX receives diverse inputs from tangential neurons, supposedly including explicit angular velocity information [11]. As there are no data indicating which specific tangential neurons assume this role in the locust, we included abstract, functionally inspired angular velocity neurons in our model. These units are designed to summarise the characteristics of biologically identified tangential neurons, potentially delivering inputs to the PB, CBL, and the lower unit of the NO (NOL). We modelled one angular velocity neuron tuned to clockwise rotation ($AV_{cw}$) and one tuned to counterclockwise rotation ($AV_{ccw}$). Their firing rates are given by

$$r_{AV_{cw}} = \frac{r_{AV_{max}}}{\frac{v_{max}}{|v|}} \cdot \mathbb{I}(v > 0),$$

$$r_{AV_{ccw}} = \frac{r_{AV_{max}}}{\frac{v_{max}}{|v|}} \cdot \mathbb{I}(v < 0),$$

(5)

where $\mathbb{I}(.)$ is the indicator function which is 1 if the argument is 'true' and 0 otherwise. $r_{AV_{max}} = 30$ Hz is the maximum firing rate of angular velocity neurons, which corresponds to a high firing rate of CX-neurons in the locust under experimental conditions [35]. $v$ is the rotational velocity, and based on data from flying locusts responding to striped patterns moving at up to 90 ˚/s [48], we conservatively assume $v_{max} = 150$ ˚/s is the maximum angular velocity of a locust.

CL1a-neurons encode the animal's orientation in a compass-like manner [6, 33, 34]. The preferred heading directions of the 16 CL1a-neurons included in our model are $\vec{\phi}_{pref} = \{k \cdot 22.5°|k = 0, \ldots, 15\}$. Here, k corresponds to the PB column index a neuron arborises in. PB columns are indexed from left to right, i.e., L8 has index 0 and R8 has index 15 (cf. Fig 2A and 2B for labelling of PB columns). The distribution of preferred heading directions of CL1a-neurons along the PB is based on the distribution of preferred solar azimuths derived from sky polarisation tuning along the PB in four types of CX neurons (CL1-, TB1-, CPU1-, and CPU2-neurons in locusts—E-PG-, Δ7-, PFL1/3-, and PFL2-neurons in fruit flies) [34]. This results in a 360˚ representation of space with a single activity maximum or compass bump along the PB (cf. Fig 1A, right panel, for a schematic representation). The firing rates of CL1a-neurons are initialised with a sinusoidal relationship to the initial heading $\phi(t_0)$ at time $t_0$:

$$r_{CL1a(t_0)} = a \cdot \cos\left(\vec{\phi}_{pref} - \phi(t_0)\right) + b,$$

(6)

where the rate amplitude $a = 5$ Hz and the operating point $b = 25$ Hz were determined from the data reported in [35].

CL2-neurons inherit heading information from CL1a-neurons. In addition, they are sensitive to rotational optic flow compatible with yaw rotation [35]. CL2-neurons and homologous neurons in the fruit fly, P-EN-neurons [2, 32], show opposite directional selectivity with neurons in the left PB hemisphere preferring counterclockwise rotations and neurons in the right PB hemisphere preferring clockwise rotations. In our model, angular velocity and direction information enter through the angular velocity neurons. They modulate the weights of synapses between CL1a- and CL2-neurons such that their firing rates change in an angular velocity-dependent manner.

To maintain a stable operating point b (see Eq 6), all CL1a- and CL2-neurons receive an additional input from a bias neuron constantly firing at ca. 100 Hz.

**Neuronal projections and connectivity assumptions.** The heading circuit connectivity was derived from anatomical projection data, based on two assumptions: first, smooth fibre

endings indicate input regions of CX-neurons and varicose fibre endings indicate output regions; second, overlapping arbors with opposite polarity are potentially synaptically connected. See Fig 2A and 2B for the general projection schemes of the modelled neuron types. The connectivity implications that follow from these two assumptions are detailed below, together with the respective evidence.

First, CL1a-neurons provide input to CL2-neurons in the PB. Each CL1a-neuron has varicose endings in a single PB column, and each CL2-neuron has smooth endings in a single PB column [1], such that each CL1a-neuron could provide input to the CL2-neuron arborising in the same column of the PB. In the fruit fly, E-PG-neurons also provide input to P-EN-neurons in the PB [2].

Second, CL2-neurons provide input to CL1a-neurons in the CBL. Each CL2-neuron has varicose endings in a single CBL column. Each CL1a-neuron has central bleb-like endings in a single CBL column surrounded by smooth endings in up to two columns on both sides of it ('mixed' endings in Fig 2A) [1]. Both CL1a- and CL2-neurons connect columns of the PB to columns of the CBL, and both neuron types project columns in each half of the PB onto alternating columns across the entire width of the CBL. The projection schemes of the two neuron types are shifted by one column (cf. the pattern of alternating light and dark gray projections across the columns of the CBL in Fig 2A and 2B). Each CL2-neuron could thus provide input to CL1a-neurons arborising in the ipsilateral as well as in the contralateral half of the PB but in the same column of the CBL. In the fruit fly, P-EN-neurons provide input to E-PG-neurons in the EB [2]. However, neuron projections differ significantly between the two species, with each E-PG-neuron innervating one of 16 wedges of the EB (corresponding to one column of the CBL) and each P-EN-neuron innervating one of eight tiles of the EB (corresponding to two neighbouring columns of the CBL).

Third, neighbouring CL1a-neurons make synaptic contacts in the CBL. The organization of varicose terminals in a single column of the CBL flanked by smooth endings in neighbouring columns renders it likely that CL1a-neurons in adjacent PB columns are synaptically connected in the CBL. Due to their projection schemes detailed above, each CL1a-neuron could provide input to other CL1a-neurons arborising in both the ipsilateral and the contralateral half of the PB. Synaptic contacts between E-PG-neurons have also been demonstrated in the EB of the fruit fly [3, 49].

Fourth, CL2-neurons arborising in the same NOL provide input to each other. In addition to the PB and the CBL, CL2-neurons arborise in the contralateral NOL. In the fruit fly, all P-EN-neurons from one hemisphere of the PB are connected with each other in the NO [3, 49], and we assume the same is true for CL2-neurons from one hemisphere in the NOL.

Lastly, angular velocity neurons potentially modulate all synapses among CL1a- and CL2-neurons. Tangential neurons provide inputs to the CX from various other brain regions [8], and many types of tangential neurons have been immunostained for neuromodulatory transmitters. Specifically, TB6-/TB7-neurons innervating the PB have been immunostained for tyramine and the neuropeptides orcokinin and locustatachykinin [50–52]. TB8-neurons (also innervating the PB) have been immunostained for octopamine [51]. TL1- and certain TL4-neurons innervating the CBL have been immunostained for the neuropeptide orcokinin [52], and TN-neurons innervating the NO have been immunostained for tyramine [51]. In our model, we summarised these properties in angular velocity model-neurons, enabling them to up- and down-regulate synapses among columnar neurons.

Fig 2C shows the resulting connectivity matrix mask for all potential synapses among CL1a- and CL2-neurons in the model. There are no data indicating the inhibitory or excitatory nature of these proposed synapses. Since data on their strength is not available either, we

determined all synaptic weights using an optimisation algorithm (see section Free parameters and their optimisation).

**Stimuli.**　We supplied two types of input stimuli to the neural circuit; heading and angular velocity. The heading stimulus was provided only once, to initialise network activity at the beginning of each experiment. The initial firing rates of the ensemble of CL1a-neurons as well as the ensemble of CL2-neurons were set to encode a particular heading direction via Eq 6. This choice of identical CL1a- and CL2-activities was motivated by the observation that P-EN and E-PG activity maxima align if the angular velocity is very low [2]. Angular velocity inputs were provided to the angular velocity neurons throughout simulations and were used to continuously update the heading signal encoded in the CL1a-neuron population activity. The angular velocity neurons initially fired at a rate corresponding to an angular velocity $v(t_0) = 0$ °/s. The initial states of all synapses in the network were set to a stationary state that is reached when the circuit receives zero angular velocity inputs for a long time (i.e. much longer than $\tau_m$ and $\tau_s$).

## Free parameters and their optimisation

The free parameters of the model were the synaptic weights (cf. Eqs 1 and 2) of all potential synapses (indicated by the connectivity mask, see Fig 2C), including both the feed-forward synapses between CL1a- and CL2-neurons and the modulatory effects of angular velocity neurons on these synapses. These parameters were determined via unconstrained gradient-based optimisation with the L-BFGS [53] algorithm. Gradients were computed with PyTorch's automatic differentiation algorithm. All synaptic weights were optimised so that the network reproduced activity targets encoding the true heading during or after an angular velocity integration time interval of 200 ms. In line with observed behaviour in the fruit fly head direction circuit reported by [54], these activity targets were defined to promote a stable heading representation with no drift when the angular velocity is zero. We used a 4th order Runge-Kutta integrator [55] to integrate the system of ordinary differential equations. Integration time steps from 1 to 8 ms yielded comparable results. We used 4 ms integration time steps for the optimisation of free parameters and 1 ms for all simulations reported here.

For a random initial heading direction $\phi(t_0)$, we computed the true heading at time $t_n$, $\phi(t_n)$, by integrating the angular velocity $v(t_n)$:

$$\phi(t_n) = \int_0^{t_n} v(t)dt + \phi(t_0). \tag{7}$$

To train the network to maintain a stable heading encoding when $v(t) = 0$ °/s (const.) throughout the integration time interval, we generated one maintenance activity target $r_{\hat{CL1a_m}} = r_{\hat{CL2_m}}$ per random heading direction. The optimisation algorithm then minimised the mean squared error between this target and the network output at every 10th integration time step. Simultaneously, to train the network to shift the heading representation when receiving nonzero angular velocity inputs, we applied one of 64 randomly drawn constant $v(t) \in [-150$ °/s, 150 °/s] for $\frac{4}{5}^{th}$ of the integration time interval, and $v(t) = 0$ °/s for the remaining $\frac{1}{5}^{th}$ of the integration time interval afterwards. The optimisation algorithm then minimised the mean squared error between the shift activity targets $r_{\hat{CL1a_s}} = r_{\hat{CL2_s}}$ resulting from integrating the angular velocity in the final 10 integration time steps and the network outputs. This choice of identical CL1a- and CL2-targets was again motivated by the observation that P-EN and E-PG activity maxima align if the angular velocity is very low [2].

Since there is no unique solution to this optimisation problem, we regularised the minimum with a low-entropy prior. The regulariser promoted similar values for all synaptic

weights across repeated connectivity structures in the PB and the CBL, i.e., it punished variance along the diagonals of the quadrants of the connnectivity matrix. This regularization was intended to produce a visually pleasing appearance of the connectivity matrix [56]. The relative weight of the regulariser was 0.1.

In Python pseudo-code, the complete objective function used to optimise the synaptic weights is

$$
\begin{aligned}
\mathcal{L}(\vec{w}) = (R_{CL1a}[:: 10] \quad &- \quad \hat{r_{CL1a_m}})^2.mean() \\
+(R_{CL2}[:: 10] \quad &- \quad \hat{r_{CL2_m}})^2.mean() \\
+(R_{CL1a}[-10:] \quad &- \quad \hat{r_{CL1a_s}})^2.mean() \\
+(R_{CL2}[-10:] \quad &- \quad \hat{r_{CL2_s}})^2.mean()
\end{aligned}
\tag{8}
$$

where $\vec{w}$ is the vector of all free parameters, or weights (cf. Eq 2) and $R_{CL1a}$ is the matrix of the rates of the 16 CL1a-neurons at each time predicted by the model. $R_{CL1a}[:: 10]$ indicates the rates of all CL1a-neurons at every tenth integration time step, and $R_{CL1a}[-10:]$ indicates the rates at the final ten integration time steps. $\hat{r_{CL1a_m}}$ is the vector of target rates during heading maintenance and $\hat{r_{CL1a_s}}$ is the vector of target rates after shifting the heading direction with angular velocity input and likewise for the CL2 rates.

## Simulations and evaluation

**Evaluating the noise robustness of heading and angular velocity integration.** We first explored the effects of altering membrane potentials, synaptic release probabilities, and synaptic weights on the accuracy of the integration of heading and angular velocity. We varied the membrane potentials by adding zero-mean Gaussian noise with standard deviation $\sigma_U \in \{0.0, 0.1, 0.5, 1.0\}$mV at every millisecond. We sampled Beta-distributed noise for the synaptic release probabilities with the mean equal to the noise-free value, and a pseudocount $\in \{10, 100, 1000\}$ (These pseudocounts correspond to an approximate coefficient of variation $\in \{0.1, 0.01, 0.001\}$). We chose a Beta distribution because it is range-limited to [0, 1], which is important for an interpretation as a probability. Finally, we randomly perturbed synaptic weights by uniformly distributed multiplicative noise with range $\in \{0.01, 0.03, 0.05\}$ at the start of each integration trial, effectively applying noise to both modulatory and non-modulatory synaptic weights relative to the original weight strength. For each noise value, we carried out N = 2000 integration trials lasting 4000 ms each. In each trial, the circuit activity was initialised based on a random heading direction via Eq 6. The angular velocity neurons received angular velocity inputs generated from lowpass filtered Gaussian noise, to mimic the observed trajectories of walking locusts. We quantified the accuracy of the heading circuit via the average angular error between the true heading (cf. Eq 7) and the heading estimate encoded in the activity of CL1a-neurons:

$$
\frac{1}{N} \sum_{n=1}^{N} |[(\phi_n(t_N) - \phi'_n(t_N) + 180°) \bmod 360°] - 180°|
\tag{9}
$$

where $\phi(t_N)$ and $\phi'(t_N)$ are the ground truth and estimated heading directions at the final points in time of each trial. Estimated heading directions $\phi'(t_N)$ were computed as the phase of the closest fitting cosine to CL1a-neuron activity. The fit was obtained by a linear regression of the cosine values to the rates $r_{CL1a}(t_N)$ with arbitrary amplitude and baseline scaling.

**Evaluating the attractor stability of the compass states.** We further explored the ability of the circuit to converge to a stable compass-like heading encoding after a perturbation of

presynaptic rates and synaptic weights. We added Gaussian noise with a standard deviation relative to the amplitude of $\sigma_{r_{pre}} \in \{0.0, 0.1, 0.25, 0.5, 0.75, 1.0\}$ to presynaptic input to the network at the beginning of the simulation. We also randomly perturbed synaptic weights by uniformly distributed multiplicative noise with range $\in \{0.01, 0.03, 0.05\}$ at the start of each integration trial. For each value, we carried out N = 1000 integration trials lasting 100 ms each. In each trial, the circuit activity was initialised via Eq 6 based on a random heading direction. Throughout the trial, angular velocity neurons received angular velocity inputs generated from lowpass filtered Gaussian noise. We quantified the stability of the heading circuit via the mean-squared deviation between the CL1a-neurons' activity and the best-fitting cosine at the end of each trial.

**Testing of the heading circuit in an agent simulation.** Lastly, we tested whether the heading circuit could guide locomotor behaviour. To simulate a walking locust in a simulated world, we linked the heading circuit to a circuit that produces outputs for goal-directed steering [40]. In short, the heading circuit outlined above updates an internal heading representation by integrating angular velocity information (cf. Fig 3A). We adapted the steering circuit to produce steering signals by comparing representations of the current heading and a constant goal direction (cf. Fig 3B). In each trial, we initialised the heading circuit's activity based

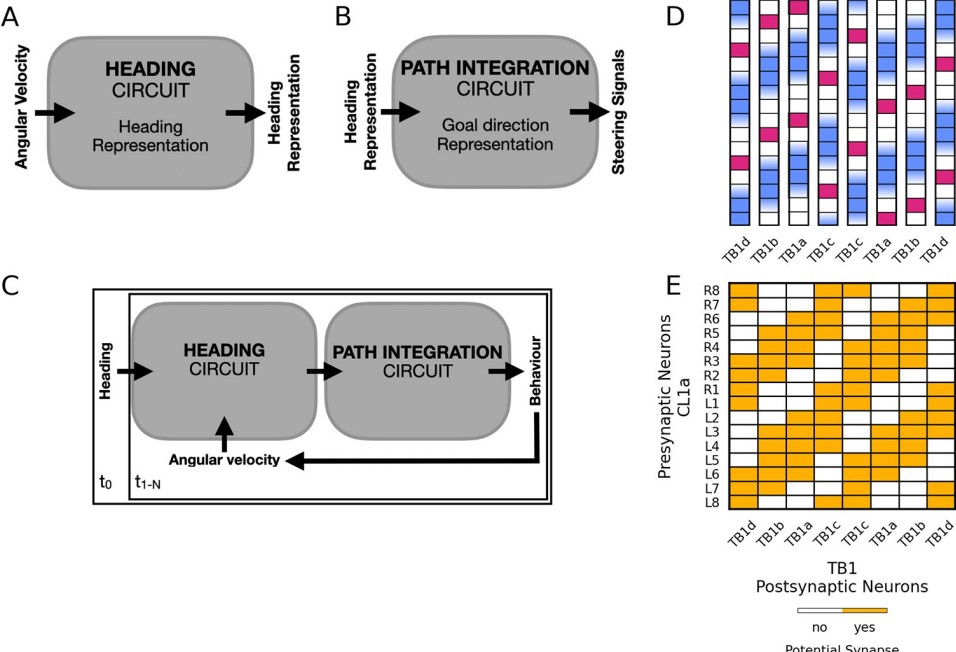

Fig 3. **Connecting the heading circuit to the goal-directed steering circuit.** (A) The heading circuit introduced above updates an internal heading representation by integrating angular velocity inputs. (B) The goal-directed steering circuit proposed by [40] produces steering signals to align the current heading with a goal direction representation. (C) The two circuits can be linked to form a closed loop. Starting with an initial heading representation (at time $t_0$), the heading representation produced by the heading circuit is used as an input to the steering circuit, which is initialised with a constant goal direction. The behavioural output of the steering circuit in turn produces angular velocity input for the heading circuit (at all consequent times $t_{1:N}$). (D) Topographic organization of TB1-neuron subtypes in the PB, adjusted from [57]. Magenta squares indicate varicose fibre endings, blue indicates smooth fibre endings. (E) Connectivity matrix mask for CL1a- and TB1-neurons within the steering circuit, adjusted for projection data from the locust illustrated in panel D (TB1-neurons) and Fig 2A (CL1a-neurons). Yellow squares indicate potential synapses without specifying excitation versus inhibition or strength. CL1a-neurons are arranged according to their position in the PB, TB1-neurons are labelled according to the subtypes with matching arborisations.

on a heading direction at time $t_0$. At all subsequent points in time, $t_{1:N}$, the updated heading representation served as an input to the steering circuit which drove a simulated motor system that moved the agent. The agent's behaviour resulted in angular velocities that were fed back into the heading circuit (cf. Fig 3C). To connect the heading circuit to the steering circuit, we had to make several adjustments to the original model of [40]: First, the CL1a output of the heading circuit was transformed via a logistic sigmoid to match the rounded square wave of CL1 activities of the goal-directed steering circuit. Second, in the original steering circuit, the current heading direction is represented by two compass bumps across 16 CL1a-neurons. This is transformed into a single bump across 8 TB1-neurons. Via optimisation, we derived a CL1a-TB1 connectivity in the steering circuit, to achieve the same with the single bump heading representation in the 16 CL1a-neurons of our model. The assumptions and optimization targets are detailed in the following: the optimization was constrained by projection data from CL1a- [1] and TB1-neurons in the locust [8, 57] (cf. Fig 3D). As outlined above, we assumed that smooth and varicose fibre endings indicate input- and output sites and that overlapping fibres with opposite polarity indicate potential synapses. Fig 3E illustrates the resulting potential synapses from CL1a- onto TB1-neurons within the steering circuit. As there are no data indicating the weights of these potential synapses, we optimised the CL1a-TB1 connections to map rounded square wave activities across 16 CL1a-neurons to similar activities across 8 TB1-neurons. The weights were constrained to be positive. Furthermore, we regularised the solution by a quadratic weight decay to push all unnecessary weights close to zero. Also, we implemented a -5° phase shift between the CL1a and TB1 bumps, to compensate for biases introduced by rounding the continuous-time representation of our heading circuit to the discrete-time steering model. This implementation-dependent bias necessitated a slightly more liberal interpretation of the CL1a-TB1 connectivity scheme depicted in Fig 3D, see Fig 4E. However, we argue that this extended connectivity is still in agreement with the data of [57] within the error margins of that data.

Third, we aimed to simulate outbound locomotor behaviour during the long-range phase of a long-distance navigational task [58]. During this phase, the animal maintains its goal direction based on global cues. In the goal-directed steering circuit, a homing vector is encoded in CPU4-neurons and updated continuously. The authors of the study introducing the goal-directed steering circuit [40] suggested that CPU4-neurons could encode a fixed direction during long-range migration in other insects, and we thus hard-coded a goal vector fixed in direction and length in this layer, instead of performing continuous updates.

The ability of the agent to maintain a steady travel direction was quantified by Eq 9, with $\phi$ and $\phi'$ as the actual and the ideal heading direction (matching the goal direction) of the agent at the final point in time of the behavioural simulation. At the beginning of each trial, the agent was placed in the simulated world with a random heading direction. This initial heading was translated to the initial heading circuit activity $r_{CL1a}(t_0)$ and $r_{CL2}(t_0)$. A random and fixed goal direction was encoded in $r_{CPU4}(t_{0:n})$. The agent then performed 200 steps, with a total duration of 20 s (an agent time step is 0.1 s long). We explored the effect of displacements by wind on the agent's performance. We modelled wind with two parameters: P(translation) is the probability of a gust of wind that displaces the agent laterally by the magnitude of one step with each step the agent takes. P(rotation) is the probability of a gust of wind that rotates the agent by a random magnitude with each step the agent takes. Each gust of wind lasted for a random duration between 500 to 1500 ms. Translation and rotation were mutually exclusive. We chose $P(translation)$, $P(rotation) \in \{0.0, 0.01, 0.02, 0.03, 0.04\}$ and repeated N = 2000 runs for each value.

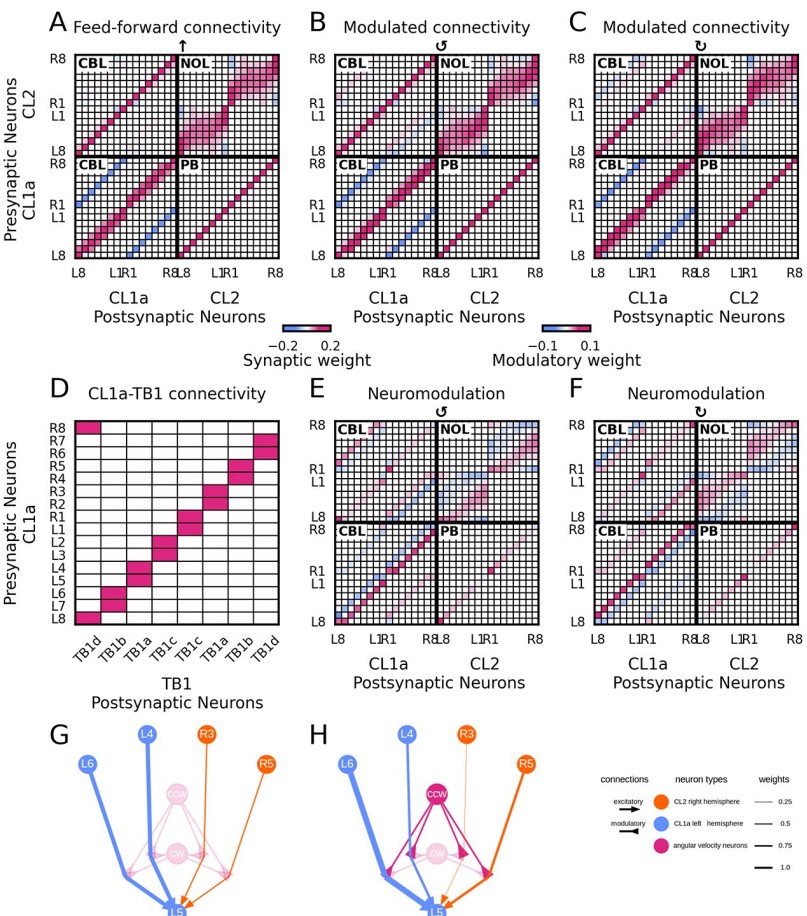

**Fig 4. Biologically constrained and optimised feedforward connectivity and neuromodulation.** (A,B,C) Effective connectivity of CL1a- and CL2-neurons. Neurons are arranged according to their position in the PB. Effective positive (excitatory) synaptic weights are displayed in magenta, negative (inhibitory) ones in blue (with the value range of the left colorbar). Colors saturate at 25% of the maximal synaptic weight value for better visibility. Panel A displays the connectivity matrix optimised to maintain a heading signal at zero angular velocity. Panels B and C illustrate the optimised, modulated circuit connectivity during counterclockwise and clockwise rotation, respectively. (E,F) Modulatory weights of angular velocity neurons onto synapses between CL1a- and CL2-neurons during counterclockwise and clockwise rotation, respectively. Positive (upregulating) modulatory weights are displayed in magenta, negative (downregulating) ones in blue (with the value range of the right colorbar). Note that B and C are the sums of A+E and A+F, respectively. (D) Optimised connectivity of CL1a- and TB1-neurons within the steering circuit. CL1a-neurons are arranged according to their position in the PB, and the eight TB1-neurons are labelled according to the subtypes with matching arborisations. Excitatory synaptic weights are displayed in magenta (with the value range of the left colorbar). (G,H) Exemplary microcircuits showing the optimised connectivity between a subset of neurons; recurrent self-connections and connections that are only weakly modulated during turns are omitted for clarity. Panel G illustrates the effective connectivity supporting activity maintenance in the heading circuit at zero angular velocity. Under these conditions, neither clockwise (cw) nor counterclockwise (ccw) angular velocity neurons are active, so synapses between CL1a- and CL2-neurons remain unmodulated. Panel H shows the modulating effect of angular velocity neurons during counterclockwise turns of the animal, which adjusts the effective connection strengths within the circuit to produce a shift in the compass bump. For a detailed description of the shift computation, please refer to the text.

# Results

## A proposed circuit for heading and angular velocity integration in the desert locust CX

The optimised circuit connectivity is illustrated in Fig 4. The CL1a-CL2 connectivity for maintaining a heading representation encoded in an activity pattern is displayed in Fig 4A. Color

saturation indicates unmodulated synaptic weight, i.e. $\tilde{g}_{s,i}$ in Eq 2. The solution features uniformly excitatory synapses from CL1a- onto CL2-neurons in the PB. In the CBL, CL1a-neurons excite CL1a-neurons arborising in the same PB arm, including excitatory self-connections, and inhibit CL1a-neurons projecting to the contralateral one. Also in the CBL, the same pattern emerges in synapses from CL2- onto CL1a-neurons, with additional inhibitory synapses between neurons arborising near the midline of the PB. In the NOL, CL2-neurons from the same hemisphere are interconnected with near excitation (including excitatory self-connections) and far inhibition. This weight matrix is symmetric, leading to a stable network activity state [59].

Modulatory effects of angular velocity neurons onto synapses between CL1a- and CL2-neurons induce a shift of the heading signal during turns. Fig 4B and 4C display the effective, modulated, connectivity ($\tilde{g}_{tot,i}$ in Eq 2) during counterclockwise and clockwise rotations. This connectivity results from adding modulatory weights $\tilde{g}_{mod,i,j}$ depicted in Fig 4E and 4F to the connectivity shown in Fig 4A, after scaling them with the pre-synaptic release probabilities $P_{rel,i,j}$. The solution features neuromodulation of synapses between all neuron populations in the PB, in the CBL, and the NOL. The core mechanism driving the shift of the activity bump during turns relies on neuromodulation of synapses between neurons arborizing in neighboring columns of the PB, as opposed to the feed-forward mechanisms found in the *Drosophila* heading system. The weight modulations during clockwise and counterclockwise turns are complementary—negative modulatory weights in Fig 4E are positive in Fig 4F and vice versa. These modulations, which are offset from the diagonals, induce asymmetries in the effective, modulated weight matrices (see Fig 4B and 4C). This directional bias allows excitation to propagate toward neighboring columns in one direction versus the other, shifting the activity bump in tune with the animal's turn direction.

This solution requires angular velocity neurons to exert both up- and down-regulating effects. Potential biological substrates for this dual effect are addressed in the discussion. The direction (up- or down-regulation) of modulatory effects is homogeneous across neuron populations (given by neuron type and hemisphere), but strengths vary. As for synapses from CL2-neurons onto CL1a-neurons and for synapses from CL1a-neurons onto CL2-neurons, modulation is more pronounced at synapses between neurons arborising in the outer- and innermost columns of the PB. A comparison of the effective connectivity at zero (Fig 4A) and nonzero angular velocity (see Fig 4B and 4C) reveals that modulation does not substantially alter overall excitation and inhibition in the circuit since modulatory effects are comparably weak.

The optimised CL1a-TB1 connectivity is shown in Fig 4D. Note that the connections far from the diagonal have been pruned, even though they would have been permissible (cf. Fig 3E). This means that only one of the two input domains of each TB1-neuron (cf. blue squares in Fig 3D) is functionally connected to a CL1a-neuron. Whether this solution emerges from the noise-free CL1a activities used for the optimisation, the positivity constraint on the weights, or the weak quadratic regulariser is a question for future investigations.

Fig 4G and 4H show exemplary microcircuits of the effective connectivity between a subset of neurons in the heading circuit at zero angular velocity and during counterclockwise turns, respectively. During turns, angular velocity neurons modulate the strengths of connections between CL1a- and CL2-neurons (see also Fig 4E and 4F).

The interplay of up- and down-regulation around the main diagonals of the connectivity matrix (in other words, the modulation of synapses between neurons in adjacent PB columns) serves a computational purpose, effectively yielding a discretised derivative of the sinusoidal CL1a activity pattern across the PB. The derivative of a sine wave is a cosine wave (and vice

versa). Any phase-shifted (co)sine wave can be computed by a weighted linear superposition of a sine and a cosine wave. This mathematical relationship is expressed by the trigonometric identity [60]:

$$\cos(k \cdot x - \Delta\phi) = a \cdot \cos(k \cdot x) + b \cdot \sin(k \cdot x) \tag{10}$$

Here, $a = \cos(\Delta\phi)$, $b = \sin(\Delta\phi)$, $k = \frac{2\pi}{16}$ and $x \in \{0, \ldots, 15\}$ indexes PB columns from L8 to R8 (labelling as depicted in Fig 2A and 2B). Consequently, the connectivity modulation introduces a linear combination of a sine and a cosine wave, resulting in a shift of the compass bump.

## Accuracy and robustness of heading and angular velocity integration

The model was evaluated in three different simulations. We first assessed the capability of the heading circuit for updating an initial heading representation by integrating a time series of angular velocity inputs (see Fig 5A for an example). Throughout these simulations, both CL1a- and CL2-neurons consistently exhibited sinusoidal activity patterns, localising in a single maximum along the PB. The position of this compass bump aligned with the ground truth heading direction (Eq 7) and dynamically responded to angular velocity inputs. As in the model of the locust heading circuit proposed by [36], the compass bump demonstrated the ability to seamlessly transition between the lateral ends of the PB. To determine how much neuromodulation in the PB, CBL and/or NOL contribute to the accuracy of angular velocity integration, we restricted modulatory inputs to either the CL1a-neurons or the CL2-neurons and re-optimized the network. We evaluated the average absolute integration error and its standard error at the

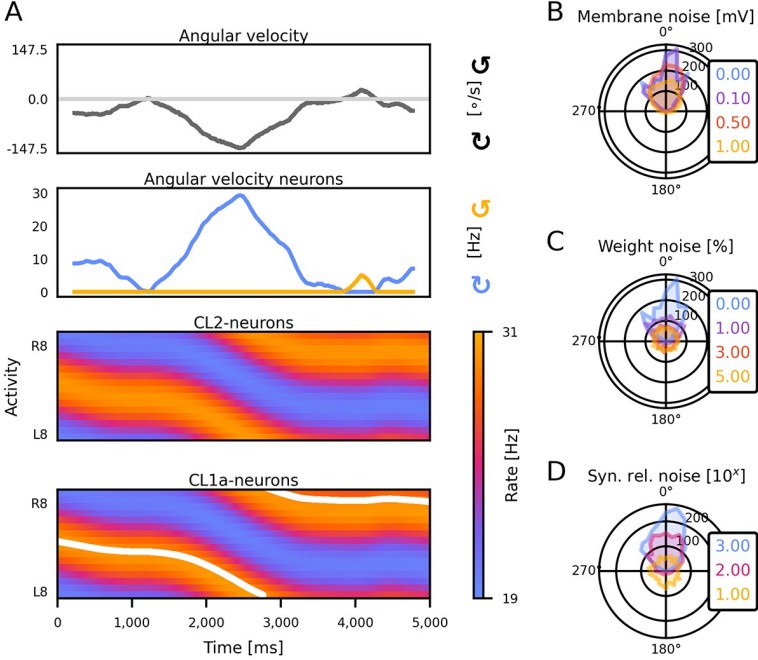

**Fig 5. The circuit integrates angular velocity signals to update the heading representation and is resilient to noise.**
(A) Activity of all neurons of the heading circuit during a noise-free trial of heading- and angular velocity integration. CL1a- and CL2-neurons are organised according to their position in the PB, revealing a single activity bump along the PB. The white line indicates the position of the ideal CL1a compass bump corresponding to the ground truth heading direction computed via Eq 7. (B-D) Accuracy of integration under increasing levels of noise. The angular deviation between the heading encoded by CL1a-neurons and the true heading is depicted (histograms from 2000 trials).

simulation endpoint after 2000 trials of 4 s duration each. Errors were 25.05˚ ± 0.39˚ for the CL1a+CL2 modulated network, 26.83˚ ± 0.45˚ for modulation of CL1a inputs only, and 26.91˚ ± 0.39˚ for a network with modulation of the CL2 inputs only. These average errors indicate that neuromodulation of either or both neuron populations will allow for comparably accurate angular velocity integration, with a slightly higher accuracy for the CL1a+CL2 modulated network.

To gauge the robustness of the CL1a+CL2 modulated circuit's integration capability, we subjected it to perturbations in three model parameters: membrane potentials, synaptic release probabilities, and synaptic weights. The circuit exhibited graceful degradation [61] in integration accuracy with increasing membrane noise. Membrane noise up to 1 mV only marginally increased the number of larger errors in the heading estimate (cf. Fig 5B). In contrast, even small amounts of weight noise have a detrimental effect on integration performance (Fig 5C). The model further demonstrated robustness to noise in the post-synaptic channel opening probability (Fig 5D). Histogram legend shows pseudocount of noise-generating Beta distribution. Except for very small pseudocounts, i.e., for high probabilities of nonzero noise, the final heading representation pointed in the right direction.

We further investigated the attractiveness of activity states. Fig 6A shows the circuit's ability to integrate the initial heading encoding and angular velocity inputs under noise-free conditions, resulting in a minimally phase-shifted copy of the initial heading encoding. An illustrative simulation with noise applied to the initial presynaptic inputs producing the initial heading representation is depicted in Fig 6B. Here, the initially noisy heading signal stabilised into an almost ideal sinusoidal activity pattern, signifying the emergence of ring attractor behaviour. Fig 6C displays an example of noise added to synaptic weights, leading to a final CL1a activity state significantly deviating from the initial heading representation or any phase-shifted variant. Given sufficient time, the circuit demonstrated resilience in balancing out noisy initial states caused by perturbations in input rates (cf. Fig 6D). However, the circuit exhibited reduced robustness and accumulated errors when subjected to perturbations in its

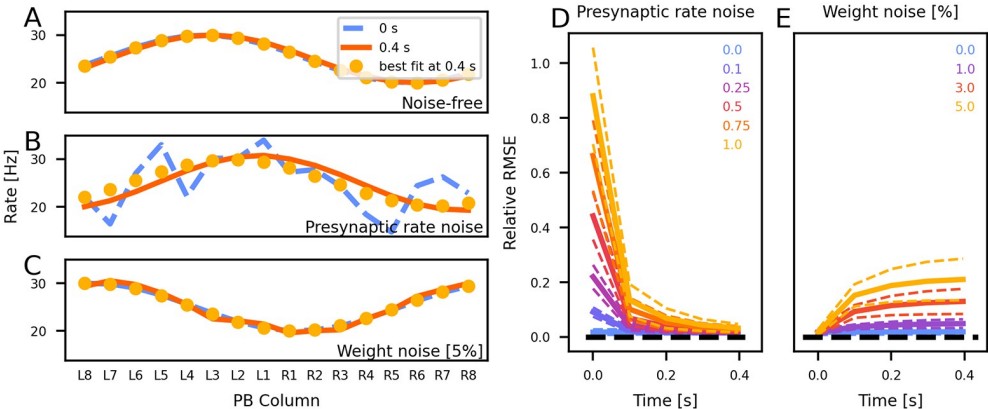

**Fig 6. The circuit maintains a sinusoidal activity pattern across the PB.** (A-C) show the initial CL1a activity at the beginning of a trial (blue) and the final activity after 0.4 s (magenta). Yellow circles indicate the sine wave best fitting to the final activity. Neurons are arranged according to their position in the PB. Panel A shows an example trial with no added noise, panel B depicts a trial with added presynaptic rate noise, and panel C shows a trial with added weight noise. (D-E) Development of the relative mean-squared error (RMSE) between the activity of CL1a-neurons at the end of each trial and the best-fitting sine (mean ± SD from 2000 trials). Panel D shows the network converging from noisy initial CL1a activity patterns to less noisy states. Panel E shows that the network activity gets more noisy with added weight noise.

connectivity induced by noise in synaptic weights (cf. Fig 6E), emphasising the optimality of the optimised circuit connectivity.

Lastly, we conducted agent simulations to gauge the heading circuit's efficacy in guiding locomotion in a predetermined goal direction. Fig 4D shows our CL1a-to-TB1 connectivity in the goal-directed steering circuit, modified from the connectivity reported by [40]. In this solution, each TB1-neuron receives excitatory inputs from two adjacent CL1a-neurons, not from one CL1a-neuron from each hemisphere as in the original model (cf. [40] S5 Fig (B)). The simulations explored the agent's ability to maintain a fixed goal direction for 20 s. Fig 7A and 7B show example trials without and with added wind perturbations, respectively. Each trial started with the agent executing a turning manoeuvre to align its heading with the goal direction. Note that the agent lacks the ability to rotate on the spot or to move sideways. In Fig 7A, the agent's heading did not improve after an initial approximate alignment with the goal direction. This may be due to a residual heading error that is too small to be corrected. In contrast, Fig 7B shows how the introduction of wind perturbations provides more pronounced feedback that helps the agent adjust its heading direction more effectively towards the goal. Upon being displaced by a wind gust, the agent resumed its previous heading. Note that the agent is not equipped with wind sensors and regulates its movements simply by comparing its current heading to the fixed goal heading. The agent exhibited robust performance across diverse probabilities of external translations and rotations, effectively balancing out the effects of perturbations. Fig 7C and 7D illustrate errors in the heading direction at the end of each simulation. The distribution of angular deviations is bimodal, indicating that the system frequently experiences minor deviations to the left and right from the goal direction but reaches exact zero deviation less often. S1(A) and S1(B) Fig demonstrate the ability of the agent to maintain straight-line orientation under conditions with different probabilities of being translated (A) or rotated (B) by wind. These panels demonstrate that left and right errors occur equally often and that overall error decreases over time, suggesting that these deviations tend to cancel each other out if they don't systematically accumulate in one direction. Importantly, these minor

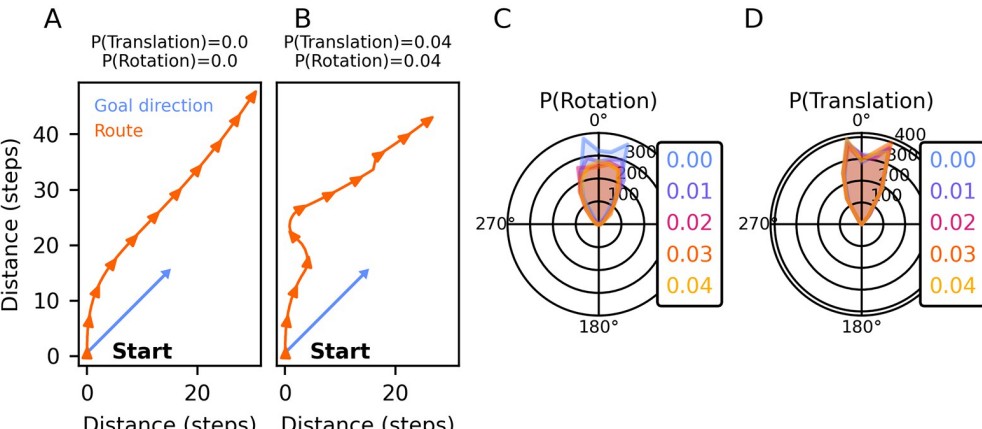

**Fig 7. Combined, the heading circuit and the adapted goal-directed steering circuit robustly guide motion in a goal direction, also in the presence of perturbations.** (A-B) Example traces illustrating the agent's motion from a starting location. The agent aligns its initial heading with the goal direction and moves in that direction. Panel A depicts a scenario without external perturbations, while panel B shows a simulation where gusts of wind cause translation and rotation of the agent. (C-D) Distribution of angular deviations between the goal heading and the agent's actual heading after 20 s (histograms from 2000 trials) under increasing probabilities of translations and rotations of the agent. In all simulations, an agent time step is 0.1 s long, and the heading circuit is integrated in 1 ms increments.

deviations are part of the system's adaptation to dynamic conditions and do not significantly affect the agent's overall ability to maintain goal-directed locomotion. Together, these results demonstrate the simulated agent's capability to carry out goal-directed locomotion in the presence of environmental perturbations. Notably, rotational perturbations can improve straight-line navigation: the bimodal error distribution in Fig 7 becomes more unimodal with increasing rotational perturbations. This is likely due to the larger error signals that are available under rotationally perturbed conditions compared to either unperturbed conditions or translation-only perturbations.

The comprehensive evaluations conducted on the proposed circuit consistently demonstrate its robust and reliable performance. While instances of failure emerged under extreme conditions surpassing the network's noise tolerance, the model exhibits remarkable resilience against minor variations in any parameter or state variable, except for noise added to its connectivity, which emerges as a pivotal network property. These findings underscore the circuit's potential for stable and accurate functioning in diverse environmental conditions, and they highlight the effectiveness of the connectivity solution obtained through optimisation.

## Discussion

Expanding on earlier linear models [35, 44], our study introduces a dynamical synapse firing rate model of angular velocity integration in a locust heading circuit. In contrast to analogous fruit fly models, this novel model exhibits a different compass topography and relies on neuromodulatory, rather than feed-forward, angular velocity inputs.

Our work is situated within the broader context of insect heading circuits, drawing inspiration from models in the fruit fly [2, 32, 62] and other insects [41]. Comparative modeling elucidates adaptations of navigation circuits to species-specific demands, contributing to our broader understanding of adaptive neural circuits. A recent comparative modeling study analysed structural differences between heading circuits in the fruit fly and desert locust [36] but did not account for data suggesting a striking functional difference. Notably, while the fruit fly circuit involves a $2 \times 360°$ compass mapping along the PB, our model assumes a single $360°$ heading representation. This distinction is implied by physiological data revealing the preferred heading directions of individual CX cells in stationary animals [6, 33, 34, 63]. It remains to be seen whether this fixed $360°$ topography is preserved during active walking or flight. Our study aims to serve as a proof of concept, exploring whether such a topography can produce a stable compass signal.

### Model constraints and properties

The model comprises CL1a- and CL2-neurons along with an abstract class of angular velocity neurons. It was constrained by morphological and functional data. Under-constrained connectivity parameters were derived via optimisation.

Heading direction is encoded in the phase of a cosine wave across the PB in our model. The notion of (co)sine waves across neural populations as representations of heading has been a longstanding topic in the literature [64]. Recent advancements, particularly the work by Aceituno and colleagues [65], have underscored their optimality under fairly general conditions, particularly concerning robustness against noise. Our model successfully updates a heading representation by integrating angular velocity signals and is robust to noise. The fruit fly model by [2] posits asymmetric feed-forward angular velocity inputs to the two halves of the PB that additively modulate firing rates in the heading circuit. Instead of relying on such feed-forward inputs, our model features neuromodulatory angular velocity inputs that effectively phase-shift the heading signal through multiplicative modulation of firing rates [43], cf. Eq 10.

While our model focuses on synaptic-level neuromodulation, other mechanisms could also achieve multiplicative effects. For instance, multiplication can occur through exponentiation by active membrane conductances of summed logarithmic inputs, nonlinear dendritic interactions, or network dynamics [66]. An anatomical offset between the projections of E-PG- and P-EN-neurons in the PB and EB of *Drosophila* has been proposed to facilitate activity bump shifts across these regions in both theoretical and computational models [2, 32]. A similar offset is conserved across species [37] and is mirrored in our circuit's connectivity (see Fig 2 for the CL1a and CL2 projection schemes and their derived connectivity). However, because our model employs single-compartment neurons, we do not explicitly separate neuron endings in different neuropils of the CX. Exploring this anatomical distinction and its effects on compass bump dynamics will be a key focus for future modeling efforts.

To complement the holistic description of the compass bump shift (Eq 10) with a circuit-level explanation, consider the microcircuits in Fig 4B and 4H, which show only the synaptic connections that are strongly modulated by angular velocity neurons during turns. We describe the bump shift from the perspective of neuron $CL1a_{L5}$ arborising in PB column L5 (cf. Fig 2A and 2B for labelling of PB columns). During straightforward movement (or standing still), this neuron receives approximately equally strong excitatory inputs from its left and right neighbors, $CL1a_{L6}$ and $CL1a_{L4}$, respectively (cf. Fig 4G). Thus, the total input received by $CL1a_{L5}$ from its neighbours is an average of its own activity, and this balance stabilizes the compass activity against noise.

When the animal turns counterclockwise, the connection to $CL1a_{L6}$ is up-regulated, and the connection to $CL1a_{L4}$ is down-regulated (see Fig 4H). If $CL1a_{L5}$ is on the rising flank of the sinusoidal compass bump, where $CL1a_{L6}$ fires less than $CL1a_{L4}$, the net input to $CL1a_{L5}$ decreases, reducing its firing rate. This reduction is necessary to shift the activity bump to the right. If $CL1a_{L5}$ is on the falling flank of the bump, where $CL1a_{L6}$ fires more strongly than $CL1a_{L4}$, strengthening the connection from $CL1a_{L6}$ to $CL1a_{L5}$ increases the net input to $CL1a_{L5}$, raising its firing rate. When $CL1a_{L5}$ is on the falling flank, this increase is needed for the bump shift to the right. The modulated connections from CL2-neurons $CL2_{R3}$ and $CL2_{R5}$ to $CL1a_{L5}$ further amplify this effect, supporting the coordinated action of CL1a-neurons $CL1a_{L4}$ and $CL1a_{L6}$ in achieving the bump shift.

In contrast to the fruit fly CX, the locust CX lacks a ring-shaped structure. However, our proposed circuit exhibits key functional ring attractor properties as described in the fruit fly [2, 32, 67]. Our model shares these properties with a previous model of the locust heading circuit featuring a $2 \times 360°$ compass representation [36]. These properties include the localisation of input to a single maximum, flexible and cyclic movement of this maximum along the attractor space, and sustained activity in the absence of input. Despite structural deviations from the fruit fly model, the shared ring attractor dynamics suggest convergent solutions to navigation tasks.

Our model of the locust heading circuit could be refined by including recurrent connections from TB1- or TB2-neurons (notice that TB1-neurons are currently not included in the heading circuit itself but appear at its interface with the goal-directed steering circuit). Whether our proposed 360° heading representation would still emerge with these recurrent connections remains to be seen. Franconville et al. [12] reported that connections from E-PG- onto P-EN-neurons in the PB are mediated by Δ7-neurons rather than being mono-synaptic. Similarly, [62] attribute the same intermediary role to Δ7-neurons (referred to as bridge neurons by [9]) in their model of the fruit fly PB. In the locust, TB1- and TB2-neurons may have a similar intermediary function. Notably, TB1a- and TB2-neurons innervate the innermost columns of the PB [57, 68] (cf. Fig 3D). Their role should be investigated in light of new evidence

suggesting that the innermost columns in the locust PB consist of two hemi-columns with divergent projection patterns [69].

## Neurobiological plausibility

The effective connectivity of our circuit model is an abstraction of the biological substrate. First, it was regularized by a simple assumption which we made in the absence of detailed biological data: anatomically similar structures might perform similar physiological functions. Second, neurons in the model can have both excitatory and inhibitory effects on other neurons. Biological implementations could feature additional interneurons, or neurons with opposing effects within the same column of the PB. Electron microscopy studies such as those by [70] could reassess the model's connectivity. Likewise, the two angular velocity units included in our model, one tuned to clockwise and one tuned to counterclockwise turns, exert up- as well as down-regulating effects on the circuit connectivity through the release of neuromodulators. These modulations may require more complex biological mechanisms than our model directly represents. For instance, it is plausible that different CL2-to-CL1a connections from the same presynaptic neuron could be modulated differently during the same turn, a process that might involve distinct neuromodulators or receptor-specific effects. This complexity could be achieved through spatially segregated synapses with different receptor expressions, or through co-transmission of neuromodulators with opposing actions, depending on the context and intracellular signaling cascades. Such effects could biologically be implemented in tangential neurons innervating the PB (TB3-TB8-neurons), CBL (TL-neurons), and the lower unit of the NO (NOL, TNL-neurons), most likely through processes more complex than those described in our model. For example, additional interneurons could effectively turn up- into down-regulation. Another explanation could be that the up- or down-regulating effect of the same modulators depends on postsynaptic receptors or signal cascades. While the presence of neuromodulatory transmitters suggests potential for modulation, their effectiveness in our model relies on more than just their presence. Specifically, for the model to work as intended, these transmitters must be able to both up- and down-regulate synaptic strengths with appropriate timing. Long-lasting neuropeptides, for instance, may not be able to provide the rapid and reversible modulation required. Future experiments should focus on determining whether the temporal dynamics of neurotransmitter release and modulation align with the requirements of our model. An additional possibility is the co-transmission [71, 72] of up- and down-regulating neuromodulators. This assumption contradicts the classical view that each neuron releases a single neurotransmitter, leading to the "one neuron, one transmitter" hypothesis [73], coined as "Dale's Principle" by [74]. However, many neurons are capable of releasing multiple neurotransmitters [75–78], and this may also be the case in the locust CX. To validate the general concept of tangential neurons acting as angular velocity neurons modulating the circuit connectivity, functional studies could assess whether TB-, TL- and TNL-neurons indeed respond to rotation cues and whether they have modulatory effects on columnar neurons. In fruit flies, GLNO-neurons have been identified as providers of rotational velocity information to P-EN-neurons, prioritizing motor signals while using visual information only under specific conditions [13]. Investigating whether analogous TNL-neurons in locusts [8, 11, 51] perform similar functions could enhance our understanding of the evolutionary conservation of these circuits across insect species.

Furthermore, the recurrent connectivity (see Fig 4A) and the attractor simulations indicate a low-pass filtering of the activity of the CL1a-neuron population. This could be tested in simultaneous neuro-stimulation and multicellular recording experiments, by injecting a noisy activity state into the PB and recording its development in time.

Regarding the employed neuron model, the use of steady-state firing rate neurons is an abstraction, and future studies need to verify that the model's basic principles of heading and angular velocity integration carry over to an implementation with spiking neurons. We have further assumed the same integration and firing dynamics in all neurons, which is likely an oversimplification and might be refined in a more comprehensive CX model. The dynamics of the neurons modelled in our study were constrained by data from homologous neurons in the fruit fly *Drosophila melanogaster*[2]. Obtaining corresponding data from the locust is crucial to explore potential differences in integration and firing dynamics specific to this species. Testing whether a lead-lag relationship between the activity maxima of E-PG- and P-EN-neurons as reported by [2] also manifests itself in CL1a- and CL2-neurons could be done with multi-compartmental models to capture the action potential transmission time along neurites.

## Simulation of goal-directed locomotion

In order to simulate locomotor behaviour, we supplied the heading representation of our model to a goal-directed steering circuit [40]. In this model, a homing vector is encoded in CPU4-neurons and constantly updated. We encoded a fixed goal direction in the CPU4-neuron population to produce steering behaviour as it would be expected during long distance migration, but other cell types are also possible candidates. In the monarch butterfly, goal direction neurons have recently been discovered [79] but were not identified morphologically. However, they could be similar to FC2- and PFL3-neurons in the fruit fly FB (corresponding to CU- and CPU1-neurons in the locust CBU) shown to encode goal directions [5].

In conjunction with the modified goal-directed steering circuit [40], our model can make behavioural predictions. Comparing reaction times of freely moving locusts to shifting visual targets in virtual reality experiments would allow deriving bounds on the functional synaptic delays in our circuit model. Furthermore, such data would allow for the comparison of our model's feedback control strategy [80] with locust behaviour.

The behavioural simulations conducted in this study are inspired by the initial phase of the proposed three phases of long-range navigational tasks [58]. During this phase, the animal maintains a steady travel direction, guided by global cues. The subsequent short-distance and pinpointing-the-goal phases rely on increasingly specific local cues, underscoring the complexity of successful long-range migration. The evaluations of the simulations we conducted here show that our proposed mechanism for angular velocity integration is robust enough to update the heading signal while other inputs are lacking or are unreliable for a short while. To allow inferences about the circuit's stability during long-range migration, multimodal cues should be available throughout simulations.

Our current approach involves initializing the activity of the heading circuit based on an initial heading direction and then supplying only angular velocity inputs to update the internal heading signal. To increase the model's realism, we plan to incorporate sky compass cues into the integration process. This will make the pathway from sky inputs to an internal heading representation with a single compass bump across the PB explicit, inspired by models featuring two compass bumps across the PB [41, 81]. Our understanding of the effective fusion of multimodal cues into a stable heading signal in the desert locust could be furthered by a computational-level analysis (following the framework of [82]) of the heading circuit. This would allow comparisons with similar models and studies of other insects, such as [83] and [84], and exploring the computational principles implemented in the circuit in greater detail. Specifically, adopting an ideal observer model [85] could make the circuit's objectives under different conditions explicit. Given the potential for conflicting information between different cue modalities in experimental manipulations, simulating experiments under both naturalistic and

laboratory conditions is crucial for a comprehensive evaluation of such a model's performance.

Identifying fundamental neuronal and computational principles of orientation across different spatial and temporal scales requires future research. In particular, it should be investigated how the various phases of navigation tasks are integrated, which cues are relevant in each action space [86], and how an animal's environment, body, and neural system [87] are coupled.

## Predictions and open questions for future research

In the previous discussion, we proposed several avenues for future experimental investigations aimed at rigorously testing our model of the locust heading circuit. Below, we summarize the key predictions and open questions derived from our model and its implications:

1. **Validating the assumption of a single compass bump in the PB using whole-population imaging techniques:** Our proposed circuit predicts a single compass bump along the PB, in contrast to the dual bumps observed in fruit flies using two-photon calcium imaging of the entire population of compass neurons [32]. If such imaging techniques were available in the locust, similar experiments would show whether our assumption holds. Dual bumps would falsify our proposal. A single bump would constitute algorithmic-level (but not mechanistic) evidence in favor of our model.

2. **Investigating model connectivity via electron microscopy:** We anticipate that detailed electron microscopy studies of CL1a- and CL2-neurons as well as tangential neurons will uncover additional synaptic connections within the heading circuit of the desert locust, refining the connectivity assumptions underlying our model.

3. **Assessing temporal dynamics alignment:** Future experiments, e.g. simultaneous pre- and postsynaptic intracellular recordings, should determine whether the temporal dynamics of neuromodulation and its electrophysiological consequences align with our model. For example, is the magnitude modulation of post-synaptic potentials in line with our predictions?

4. **Examining the role of tangential neurons in angular velocity integration:** We expect physiological studies to show that TB-, TL-, or TNL-neurons respond to rotational cues, comparable to GLNO-neurons in the fruit fly [13]. Further functional studies of the modulatory neurotransmitters detected in these tangential neurons [8] could validate their roles in our model of the locust heading circuit.

5. **Examining the role of TB-neurons:** We predict that including TB1- and TB2-neurons in future models will reveal significant contributions to heading representation, enhancing our understanding of their potential role as intermediaries in the circuit.

6. **Implementing spiking neuron models:** We anticipate that spiking neuron models will replicate the principles of heading and angular velocity integration implemented in our model. This would confirm our model's applicability to biological systems and underscore neuromodulation as a potential mechanism for introducing asymmetries into heading circuits [43].

7. **Testing lead-lag relationships between CL1a and CL2 activity bumps:** Future circuit models employing multi-compartmental neurons should demonstrate reversed lead-lag relationships in the activity of CL1a- and CL2-neurons between the PB and CBL, mirroring findings from E-PG- and P-EN-neurons in the fruit fly [2].

8. **Determining synaptic delays:** We expect that experiments on reaction times to shifting visual target stimuli in locusts will yield upper bounds for our assumptions about synaptic delays. Two-photon calcium imaging studies of CL1a- and CL2-neurons in locusts, as performed with E-PG- and P-EN-neurons in fruit flies [2], would be crucial for exploring species-specific differences in integration and firing dynamics of the involved neurons.

9. **Investigating low-pass filtering dynamics:** We predict that simultaneous neuro-stimulation and multicellular recordings at different positions in the PB will indicate that the CL2-neurons' activity is a low-pass filtered version of the CL1a-neurons' activity, when the latter are stimulated. Furthermore, it will be interesting to see if the time constants of this low-pass filter are comparable to those deduced from *Drosophila* data, i.e. in the order of 100 ms (see supporting information S2 Text).

10. **Assessing multimodal cue integration:** Simulating cue integration in both naturalistic and laboratory conditions should reveal how well the model performs under conflicting cues, allowing comparisons with studies of other insect species [83, 84].

11. **Incorporating sky compass cues:** We expect that integrating sky compass cues will enhance the accuracy of the model's heading estimate and its predictive capacity for natural heading updates. This would close the loop between environment, body, and brain [87], allowing for simulations of experiments from both field and laboratory settings.

12. **Exploring integration across temporal and spatial scales:** Future research should investigate how different phases of navigation tasks are integrated, particularly focusing on which cues from a multimodal set are most relevant for orientation in each context [17, 86].

## Supporting information

**S1 Text. Neuron model derivation.**
(PDF)

**S2 Text. Functional transmission delays between CL1a- and CL2-neuron populations.**
(PDF)

**S1 Fig. Supplementary Figure to Fig 7.** Mean-squared deviation between the agent's heading estimate and the ground truth heading direction, averaged over 1000 trials lasting 20 s each, recorded every 5 s. Panels A and B demonstrate the ability of the agent to maintain straight-line orientation under conditions with different probabilities of being translated (A) or rotated (B) by wind. Solid lines show the mean angular deviation, dashed lines are mean ± one standard deviation. For details, see text.
(TIF)

**S2 Fig. Sigmoid rate function approximation.** Green dotted line: rate function of the ideal integrate-and-fire neuron without noise. Blue dashed line: rate function with $\sigma = 8$ mV membrane noise, averaged across 1000 simulations. Solid orange line: best fit obtained with a logistic sigmoid. For details, see text.
(TIF)

**S3 Fig. Two integrate-and-fire neurons $n_1$, $n_2$ connected by an excitatory synapse.** Top panel: sinusoidal input current to $n_1$. Second panel: membrane potential and spikes of $n_1$. Third panel: membrane potential and spikes of $n_1$. Bottom panel: post-synaptic open

probability $P_{n_1 \to n_2}$.
(TIF)

**S4 Fig. Comparison of integrate-and-fire neuron rates to rate neuron model predictions.**
Top panel: rate of neuron $n_1$. Histogram computed from 1000 repetitions of the integrate-and-fire simulation. Lines show rate model predictions with a Poisson synapse. Bottom panel: rate of neuron $n_2$.
(TIF)

## Author Contributions

**Conceptualization:** Kathrin Pabst, Barbara Webb, Uwe Homberg, Dominik Endres.

**Formal analysis:** Kathrin Pabst, Dominik Endres.

**Funding acquisition:** Barbara Webb, Uwe Homberg, Dominik Endres.

**Investigation:** Kathrin Pabst, Evripidis Gkanias, Dominik Endres.

**Methodology:** Kathrin Pabst, Evripidis Gkanias, Barbara Webb, Dominik Endres.

**Project administration:** Kathrin Pabst, Dominik Endres.

**Resources:** Dominik Endres.

**Software:** Kathrin Pabst, Evripidis Gkanias, Dominik Endres.

**Supervision:** Dominik Endres.

**Validation:** Dominik Endres.

**Visualization:** Kathrin Pabst, Dominik Endres.

**Writing – original draft:** Kathrin Pabst, Dominik Endres.

**Writing – review & editing:** Evripidis Gkanias, Barbara Webb, Uwe Homberg.

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
