## [Decision Letter · Decision Letter 0]

9 Aug 2024

Dear Ms Pabst,

Thank you very much for submitting your manuscript "A computational model for angular velocity integration in a locust heading circuit" for consideration at PLOS Computational Biology.

As with all papers reviewed by the journal, your manuscript was reviewed by members of the editorial board and by several independent reviewers. In light of the reviews (below this email), we would like to invite the resubmission of a significantly-revised version that takes into account the reviewers' comments.

We cannot make any decision about publication until we have seen the revised manuscript and your response to the reviewers' comments. Your revised manuscript is also likely to be sent to reviewers for further evaluation.

Sincerely,

Matthieu Louis

Academic Editor

PLOS Computational Biology

Daniele Marinazzo

Section Editor

PLOS Computational Biology

Reviewer's Responses to Questions

**Comments to the Authors:**

Reviewer #1: Review is uploaded as an attachment.

Reviewer #2: Review is uploaded as attachement.

**Have the authors made all data and (if applicable) computational code underlying the findings in their manuscript fully available?**

Reviewer #1: **No: **Unless I am mistaken, the available code is limited to data shown in the figures. Code needed to compute the parameters in equation 3, for example, does not seem to be provided. I am sure the authors will correct this if asked by the journal.

Reviewer #2: Yes

PLOS authors have the option to publish the peer review history of their article (what does this mean?). If published, this will include your full peer review and any attached files.

Reviewer #1: No

Reviewer #2: No
---

## [Decision Letter · Decision Letter 1]

25 Nov 2024

Dear Ms Pabst,

We are pleased to inform you that your manuscript 'A computational model for angular velocity integration in a locust heading circuit' has been provisionally accepted for publication in PLOS Computational Biology.

Before your manuscript can be formally accepted you will need to complete some formatting changes, which you will receive in a follow up email. A member of our team will be in touch with a set of requests. We also recommend that you try to incorporate some of the comments of reviewer #2  in the final version of your manuscript (without the need of making any major changes).

Best regards,

Matthieu Louis

Academic Editor

PLOS Computational Biology

Daniele Marinazzo

Section Editor

PLOS Computational Biology

Feilim Mac Gabhann

Editor-in-Chief

PLOS Computational Biology

Jason Papin

Editor-in-Chief

PLOS Computational Biology

Reviewer's Responses to Questions

**Comments to the Authors:**

Reviewer #1: Dear authors,

in the revised version, all my concerns have been addressed.

Best regards

Reviewer #2: Review is uploaded as an attachment.

**Have the authors made all data and (if applicable) computational code underlying the findings in their manuscript fully available?**

Reviewer #1: Yes

Reviewer #2: None

PLOS authors have the option to publish the peer review history of their article (what does this mean?). If published, this will include your full peer review and any attached files.

Reviewer #1: No

Reviewer #2: No

---

## [Editor Report · Acceptance letter]

15 Dec 2024

PCOMPBIOL-D-24-00782R1 

A computational model for angular velocity integration in a locust heading circuit

Dear Dr Pabst,

I am pleased to inform you that your manuscript has been formally accepted for publication in PLOS Computational Biology. Your manuscript is now with our production department and you will be notified of the publication date in due course.

With kind regards,

Lilla Horvath
